# Effects and Mechanisms of Chitosan and ChitosanOligosaccharide on Hepatic Lipogenesis and Lipid Peroxidation, Adipose Lipolysis, and Intestinal Lipid Absorption in Rats with High-Fat Diet-Induced Obesity

**DOI:** 10.3390/ijms22031139

**Published:** 2021-01-24

**Authors:** Shing-Hwa Liu, Rui-Yi Chen, Meng-Tsan Chiang

**Affiliations:** 1Graduate Institute of Toxicology, College of Medicine, National Taiwan University, Taipei 10051, Taiwan; shinghwaliu@ntu.edu.tw; 2Department of Medical Research, China Medical University Hospital, China Medical University, Taichung 40402, Taiwan; 3Department of Pediatrics, College of Medicine, National Taiwan University Hospital, Taipei 10051, Taiwan; 4Department of Food Science, National Taiwan Ocean University, Keelung 20224, Taiwan; v33322733@gmail.com

**Keywords:** high-molecular-weight chitosan, chitosan oligosaccharide, high-fat diet-fed rats, lipid metabolism

## Abstract

Chitosan and its derivative, chitosan oligosaccharide (CO), possess hypolipidemic and anti-obesity effects. However, it is still unclear if the mechanisms are different or similar between chitosan and CO. This study was designed to investigate and compare the effects of CO and high-molecular-weight chitosan (HC) on liver lipogenesis and lipid peroxidation, adipose lipolysis, and intestinal lipid absorption in high-fat (HF) diet-fed rats for 12 weeks. Rats were divided into four groups: normal control diet (NC), HF diet, HF diet+5% HC, and HF diet+5% CO. Both HC and CO supplementation could reduce liver lipid biosynthesis, but HC had a better effect than CO on improving liver lipid accumulation in HF diet-fed rats. The increased levels of triglyceride decreased lipolysis rate, and increased lipoprotein lipase activity in the perirenal adipose tissue of HF diet-fed rats could be significantly reversed by both HC and CO supplementation. HC, but not CO, supplementation promoted liver antioxidant enzymes glutathione peroxidase and superoxide dismutase activities and reduced liver lipid peroxidation. In the intestines, CO, but not HC, supplementation reduced lipid absorption by reducing the expression of *fabp2* and *fatp4* mRNA. These results suggest that HC and CO have different mechanisms for improving lipid metabolism in HF diet-fed rats.

## 1. Introduction

Today, obesity is now a major chronic disease. The World Health Organization (WHO) has mentioned that 39% of adults aged 18 and over in 2016 were overweight and 13% were obese; 38 million children under the age of 5 were considered to be overweight or obese in 2019 [1]. Obesity is one of the major risk factors for the occurrence of nonalcoholic fatty liver disease (NAFLD) [2,3]. Hypercholesterolemia is known as a critical risk factor for the initiation and progression of NAFLD [2,3]. How to prevent or improve obesity is an important issue to decline the NAFLD prevalence.

Chitosan is a cationic high-molecular-weight polymer in nature. It is a partially deacetylated polymer of N-acetylglucosamine derived from polysaccharide chitin and is primarily extracted from shells of crustaceans such as shrimp and crabs and from squid pens [4,5]. Due to its functional properties such as antibacterial activity, non-toxicity, easy modification and biodegradability, it is a biologically active polymer with a wide range of applications, including agriculture, food processing, water and waste treatment, cosmetics, and biopharmaceuticals [6,7]. Chitosan oligosaccharides are mainly produced by acid hydrolysis or enzyme hydrolysis of chitin or chitosan [8]. It is an oligomer of β-(1→4)-linked D-glucosamine that possesses good water solubility, biocompatibility, and safety and is easily absorbed by the body [6]. Supplementation of chitosan or chitosan oligosaccharide has been found to be capable of regulating glucose and lipid metabolisms and improving obesity or diabetes [9,10,11,12,13].

Liu et al. (2018) have compared the effects and mechanisms on lipid metabolism between low-molecular weight (MW) chitosan (LC) and high-MW chitosan (HC) supplementation in high-fat (HF) diet-fed rats for 8 weeks [14]. They found that 5% HC supplementation possessed a higher efficiency than 5% LC supplementation on the intestinal lipid absorption inhibition and an enhancement of hepatic fatty acid oxidation, leading to improvement in liver lipid biosynthesis and accumulation [14]. A previous study has also found that the effects and mechanisms on improving lipid metabolism are different between 5% chitosan oligosaccharides (CO) and 5% HC/5% LC in HF diet-fed rats for 8 weeks [15]. They found that HC or LC supplementation promoted liver AMPK and PPARα activation, which caused the reduction in lipid regeneration and the promotion of lipid β-oxidation. However, 5% CO supplementation could not improve the HF diet-increased cholesterol levels in the liver and it might induce liver damage [15]. Nevertheless, the effects and mechanisms of CO on lipid metabolism in the liver and adipose tissue are not fully understood. The comparison of therapeutic potentials on functional properties of regulating lipid metabolism and intestinal/fecal lipids between HC and CO is not well documented. Therefore, the present study focused on exploring the effects and mechanisms of CO compared to HC supplementation on lipid metabolism, including hepatic lipogenesis and lipid peroxidation, adipose tissue lipolysis, and intestinal lipid absorption, in HF diet-induced obesity in rats. We designed a longer experimental period (12 weeks) than that in the previous studies (8 weeks).

## 2. Results and Discussion

### 2.1. The Changes in Body Weight, Food Intake, Food Utilization Rate, and Tissue Weight

As shown in Table 1, the final body weight and body weight gain in HF diet-fed rats were markedly increased (*p* < 0.05 vs. normal control diet (NC) group), which could be effectively reversed by both 5% HC and 5% CO supplementation (*p* < 0.05 vs. HF diet-fed group). There was no significant difference in food intake among the HF diet-fed, HC, and CO groups (*p* > 0.05), although the food intake of both HC and CO groups was decreased compared to NC group (*p* < 0.05). The food efficiency in HF diet-fed, HC, and CO groups were higher than the NC group (*p* < 0.05), while it was lower in both HC and CO groups than that of the HF group (*p* < 0.05). These results showed that supplementation of both 5% HC and 5% CO to the HF diet-fed rats for 12 weeks significantly reduced body weight without affecting food intake. These results are consistent with the findings of Liu et al. [16] and Huang et al. [9].

The liver weight was increased and cecal weight and cecal content weight were decreased in the HF diet-fed rats (*p* < 0.05 vs. NC group), which could be reversed by HC (*p* < 0.05), but not CO, supplementation (Table 2). The perirenal, epididymal, and total adipose tissue weights were increased in the HF diet-fed rats (*p* < 0.05 vs. NC group), which could be reversed by both HC and CO supplementation (*p* < 0.05 vs. HF diet-fed group; Table 2). There was no significant difference among these four groups in the length and weight of the small intestine (*p* > 0.05).

These results showed that supplementation of 5% HC to the HF diet-fed rats for 12 weeks significantly reduced the relative weights of liver and perirenal/total adipose tissue. These results are consistent with the findings of Liu et al. [14]. Supplementation of 5% CO significantly reduced the relative weight of epididymal adipose tissue, while it had no significant effect on relative weights of liver. These results are consistent with the findings of Chiu et al. [15].

We also found that the weight of the cecal content in the HC group was significantly higher than that of the CO group. A previous study has shown that supplementation of corn bran containing high dietary fibers to the feed can increase the cecal content in rats compared to those on a dietary fiber-free diet [17], indicating that the cecal content may be affected by the dietary fibers in the diet. Chitosan is insoluble in water, so it can be regarded as an insoluble dietary fiber [18]. The absorption of chitosan in the intestine is mainly affected by its molecular weight and water solubility [19]. The molecular weight of chitosan oligosaccharides is small and the water solubility is good, and it is absorbed more easily in the intestine. Accordingly, in the present study, the weight of the cecum content was lighter in the CO group than that of the HC group.

### 2.2. The Changes in Plasma Biochemical Indexes

We next examined the changes in plasma biochemical indexes in HF diet-fed rats supplemented with 5% HC and 5% CO. As shown in Table 3, the plasma levels of total cholesterol (TC), very low-density lipoprotein cholesterol (VLDL-C), low-density lipoprotein cholesterol (LDL-C) + VLDL-C, and TC/ high-density lipoprotein cholesterol (HDL-C) were increased in the HF diet-fed rats (*p* < 0.05 vs. NC group), which could be reversed by both HC and CO supplementation (*p* < 0.05 vs. HF diet-fed group). The plasma levels of LDL-C and HDL-C/ (LDL-C+ VLDL-C) were markedly increased and decreased, respectively, in the HF diet-fed rats (*p* < 0.05 vs. NC group), which could also be reversed by HC (*p* < 0.05 vs. HF diet-fed group), but not CO, supplementation (Table 3). Both HC and CO supplementation could not reverse the decreased plasma HDL-C levels in the HF diet-fed rats (*p* > 0.05; Table 3). Unexpectedly, the plasma levels of triglyceride were markedly decreased in the HF diet-fed, HC, and CO groups (*p* < 0.05 vs. NC group; Table 3). The plasma levels of triglyceride (TG) in the HC group were lower than that of the HF diet-fed group (*p* < 0.05), while there was no significant difference between the CO group and HF diet-fed group (*p* > 0.05) (Table 3). The blood glucose levels were increased in the HF diet-fed rats (*p* < 0.05 vs. NC group), which could be reversed by both HC and CO supplementation (*p* < 0.05 vs. HF diet-fed group) (Table 3).

As shown in Figure 1, the plasma aspartate aminotransferase (AST) and alanine aminotransferase (ALT) activities were slightly, but not significantly, increased (*p* > 0.05) and significantly increased (*p* < 0.05) in the HF diet-fed rats, respectively, which could not be reversed by both HC and CO supplementation. Unexpectedly, in the CO group, the plasma ALT and AST activities were increased compared to the NC group (*p* < 0.05); the ALT activity in the CO group was even higher than that of the HF diet-fed group (*p* < 0.05).

These results showed that HC supplementation significantly reduced the levels of TC, LDL-C, VLDL-C, and TG in the plasma, but had no effect on the HDL-C level. These results are consistent with the findings of Liu et al. [16]. A previous study has indicated that 5% chitosan can promote the liver LDL receptor (LDLR) mRNA expression, increase the clearance of LDL into the liver, and reduce the plasma cholesterol level [20]. We also found that CO supplementation significantly reduced the levels of TC and VLDL-C, but not other lipids, in the plasma. Zong et al. (2012) have found that CO supplementation increases the liver scavenger receptor BI and LDLR mRNA expression, leading to an enhancement of the reverse cholesterol transport and reduction in plasma levels of TG and LDL-C [21]. Moreover, we found that the plasma TC and LDL-C levels in the HF diet group were significantly higher than the NC group, but the plasma TG level was significantly lower than the NC group. This paradoxical decrease in plasma TG levels in the HF diet group is exactly consistent with previous findings, which may be caused by the down-regulation of plasma microsomal angiopoietin-like 4 (a suppressor of lipoprotein lipase) and hepatic TG transfer protein and apolipoprotein E (a VLDL-TG secretion enhancer), leading to the TG accumulation in the liver [11,22,23]. Moreover, a previous study has also found that the rate of secretion of VLDL by hepatocytes and the release of TG from the liver are decreased when the TG levels in the liver exceed 10% [24]. These factors may cause the reduction in TG level in the plasma of the HF diet group.

### 2.3. The Changes in Liver Lipids and Biochemical Indexes

As shown in Table 4, the TC and TG levels in the livers were significantly increased in the HF diet-fed rats group (*p* < 0.05 vs. NC group), which could be significantly reversed by both HC and CO supplementation (*p* < 0.05 vs. HF diet-fed group).

Supplementation of 5% CO in HF diet-fed rats for 8 weeks has been shown to exert liver damage (AST and ALT activities increase) via a higher hepatic cholesterol accumulation and a higher intestinal cholesterol uptake [15]. However, in the present study, a 12-week feeding study of 5% CO showed different results: AST activity was still increased but there was no significant change in ALT activity (Figure 1) and liver TC and TG levels were effectively decreased (Table 4).

Both acetyl-CoA carboxylase (ACC) and fatty acid synthase (FAS) are the key enzymes for liver fatty acid synthesis. It has been shown that the levels of FAS and ACC mRNA expression in the liver are increased, leading to an increase in the liver TG levels in HF diet-fed mice [25]. Moreover, HMG-CoA reductase (HMGCR) is known to catalyze the transformation of HMG-CoA into mevaionate in the liver, which is a rate determining step of cholesterol biosynthesis. We further examined the changes in liver lipid biosynthesis enzyme activities. The activities of ACC, FAS, and HMGCR were markedly increased in the HF diet-fed rats (*p* < 0.05 vs. NC group), which could be significantly reversed by both HC and CO supplementation (*p* < 0.05 vs. HF diet-fed group) (Table 4).

The liver tissue section stained with H&E showed that the areas of fat vacuoles were increased in the HF diet-fed rats (*p* < 0.05 vs. NC group), which could be reversed by HC (*p* < 0.05 vs. HF diet-fed group), but not CO (*p* > 0.05 vs. HF diet-fed group), supplementation (Figure 2).

AMPK is a key role in regulating the balance of cell catabolism and anabolism. A previous study has indicated that AMPKα activation plays a key role in cholesterol homeostasis by inhibiting hepatic cholesterol synthesis [26]. AMPK activation can down-regulate lipogenic transcription factors PPARγ and SREBP-1c, thereby inhibiting the lipogenic genes [27,28]. Supplementation of high MW chitosan has been demonstrated to alleviate the lipogenesis in HF diet-fed rats through the mechanisms of AMPK activation and lipogenesis-associated gene inhibition [22].

We next tested the changes in AMPK and PPARγ signaling molecules in the livers. As shown in Figure 3, the levels of protein expression of phosphorylated AMPKα and PPARγ in the livers of HF diet-fed rats were significantly decreased and increased, respectively (*p* < 0.05 vs. NC group), which could be reversed by both HC and CO supplementation (*p* < 0.05 vs. HF diet-fed group). These results indicated that both HC and CO supplementation could significantly increase the activation of AMPKα, and significantly reduce the PPARγ protein expression, and then inhibit the activities of ACC, FAS, and HMGCR to reduce the levels of TC and TG in the liver.

We also investigated the changes in antioxidant enzyme activities and lipid peroxidation in the livers. As shown in Figure 4, the levels of liver thiobarbituric acid reactive substances (TBARS) and antioxidant enzyme activities of both glutathione peroxidase (GPx) and superoxide dismutase (SOD) were significantly increased and decreased, respectively (*p* < 0.05 vs. NC group), which could be reversed by HC (*p* < 0.05 vs. HF diet-fed group), but not CO (*p* > 0.05 vs. HF diet-fed group), supplementation. Moreover, the immunochemical staining of 4-hydroxy-2-nonenal (4-HNE) for lipid peroxidation in the livers showed that 4-HNE-positive areas were increased in the HF diet-fed rats (*p* < 0.05 vs. NC group), which could be reversed by HC (*p* < 0.05 vs. HF diet-fed group), but not CO (*p* > 0.05 vs. HF diet-fed group), supplementation (Figure 5).

It has been found that rats fed with an HF diet reduce the activities of GPx and glutathione-s-transferases (GST) and increase TBARS levels in the liver [29]. Wang et al. (2017) have also shown that chitosan can reduce hepatic TBARS level in HF diet-fed mice [30]. Our results showed that HC supplementation could also increase liver antioxidant enzymes GPx and SOD activities, and reduce the production of TBARS and 4-HNE in HF diet-fed rats. These results are similar to the findings of previous studies. In contrast, CO supplementation had no significant effect on the liver antioxidant enzyme activities and lipid peroxide TBARS and 4-HNE production, but it increased the AST activity. Teodoro et al. (2016) have indicated that treatment with CO for six weeks significantly increases the AST activity in the plasma of diabetic rats [31]. Zhang et al. (2019) have also found that rats intragastrically treated with 50 mg/kg CO (average MW 5000 Da; deacetylation degree (DD) > 90%) for two weeks do not change the lipid peroxidation and anti-oxidative enzyme activities in the hearts of normal control rats [32]. Our results are consistent with the findings of these previous studies.

### 2.4. The Changes in Triglyceride, Lipolysis Rate, and Lipoprotein Lipase (LPL) Activity in the Perirenal Adipose Tissue, fabp2 and fatp4 mRNA Expression in the Small Intestinal Mucosa, and Fecal Weight and TC and TG Levels in Feces

As shown in Figure 6, in the perirenal adipose tissues of HF diet-fed rats, the TG level was increased, the lipolysis rate was decreased, and the LPL activity was increased (*p* < 0.05 vs. NC group). Supplementation of both HC and CO could significantly reverse these changes in the HF diet-fed rats (*p* < 0.05 vs. HF diet-fed group).

High MW chitosan supplementation in high-fructose diet-fed rats has been suggested to increase the plasma ANGPTL4 expression, enhance lipolysis rate, and inhibit LPL activity, leading to a decrease in the adipose tissue weights [13]. Our results are consistent with the findings of this previous study.

We further investigated the changes in *fabp2* and *fatp4* mRNA expression in the small intestinal mucosa. As shown in Figure 7, the levels of *fabp2* and *fatp4* mRNA expression were significantly increased in the HF diet-fed rats. Supplementation of CO, but not HC, significantly reversed the increased *fabp2* and *fatp4* mRNA expression (*p* < 0.05 vs. HF diet-fed group; Figure 7). Unexpectedly, the *fatp4* mRNA expression was significantly increased in the HC group compared to HF diet group (*p* < 0.05; Figure 7B).

The fecal weight and lipid contents were also examined. As shown in Table 5, there was no significant difference for the changes in fecal dry and wet weights among the four different diets groups (*p* > 0.05). Supplementation of HC to an HF diet significantly increased the fecal TC and TG levels (*p* < 0.05 vs. HF diet-fed group), while CO supplementation slightly, but not significantly, increased the levels of fecal lipids in HF diet-fed rats (*p* > 0.05 vs. HF diet-fed group). Moreover, HC, but not CO, supplementation could significantly increase the amount of bile acid excretion into feces in HF diet-fed rats (*p* < 0.05 vs. HF diet-fed group).

FABP2 and FATP4 actions are significantly related to fatty acid absorption [33,34]. Our results indicated that CO, but not HC supplementation effectively inhibited the expression of intestinal *fabp2* and *fatp4* mRNA, leading to reduce the intestinal lipid absorption. On the other hand, chitosan dissolves in an acidic environment and when it is in the stomach, the positively charged amine groups on the chitosan will form emulsified micelles with negatively charged cholesterol and lipids in a diet. Because chitosan is coated on the surface of lipid droplets, intestinal enzymes cannot decompose chitosan, thereby reducing lipid absorption, which allows lipid droplets to be excreted in the form of feces along with chitosan [35]. Chitosan oligosaccharides have small molecular weight, good water solubility, and are easily absorbed, making their lipid adsorption capacity worse than chitosan. This is the reason why a significant increase in cecal weight, cecal contents, and fecal bile acids was found in obese rats fed a diet with HC supplementation but not in those fed with CO supplementation. Therefore, the physical properties of chitosan make a large amount of lipids in diet excretion with chitosan, and this action may induce a feedback increase in the expression of *fatp4* mRNA to satisfy a sufficient amount of fatty acid absorption in the intestine.

It is usually too difficult to detect the exact phase of the menstrual cycle with different plasma levels of sex hormones in female rats. Therefore, in this study, we used male rats that have proper responses. Nevertheless, it may be necessary to investigate these experiments in female to see if the same results can be extended to females in the future.

## 3. Materials and Methods

### 3.1. Materials

High-molecular-weight chitosan was obtained from Charming & Beauty Co., Ltd. (Taipei, Taiwan). Chitosan oligosaccharide was purchased from Koyo Chemical Co., Ltd. (Tokyo, Japan). The average MW of high-MW chitosan and chitosan oligosaccharide was about 740 kDa and 719 Da, respectively; the deacetylation degree (DD) of high-MW chitosan and chitosan oligosaccharide was about 91% and 100%, respectively.

### 3.2. Experimental Animals

The animal study was approved by the Animal House Management Committee of the National Taiwan Ocean University (approval number: 107038) and was conducted by the guidelines for care and use of laboratory animals [36]. Six-week-old male Sprague-Dawley (SD) rats were provided by BioLASCO Taiwan Co., Ltd. (Taipei, Taiwan). Rats were kept in cages with husbandry conditions at 23 ± 1 °C, 40–60% relative humidity, and a 12 h light/12 h dark cycle. Rats had free access to a standard laboratory diet (5001 rodent diet, LabDiet, St. Louis, MO, USA) and water was purified by reverse osmosis. Rats were randomly divided into four groups after a 1-week acclimation: normal control diet (chow diet) (NC), HF diet (chow diet + 10% lard) (HF), HF diet + 5% high MW chitosan (HC), HF diet + 5% chitosan oligosaccharides (CO). The doses for HC and CO were selected according to the previous studies [14,15,22] and our preliminary test. The composition of experimental diets was shown in Table 6. The experiment period was 12 weeks. Body weight was detected once a week and food intake was analyzed every 3 days. At the last three days of week 12, feces were collected, dried, and weighed (wet and dry feces).

### 3.3. Sampling Blood and Tissue

After 12 weeks of experimental administration, the samples of blood and tissues for liver, perirenal and para-epididymal adipose, small intestine, and cecum were collected after euthanasia of rats under anesthesia. The preparation of plasma was determined by centrifugation at 1750× *g* for 20 min (4 °C). The samples were immediately frozen and stored at −80 °C until further experiment.

### 3.4. Analysis of Plasma Lipids, Liver Lipids, and Lipoproteins

The detection of plasma TC and TG levels was performed by the enzymatic assay kits for TC and TG (Audit Diagnostics, Cork, Ireland). A spectrophotometer (UV/VIS-7800, JASCO International) was used to detect the absorbance at 500 nm.

The liver lipids were extracted as previously described by Folch et al. [37]. The detection of TG and TC levels in the livers was performed as previously described by Carlson and Goldfarb [38].

The lipoprotein cholesterol in the plasma was separated and detected as previously described by Takehisa and Suzuki [39]. The high-density lipoprotein cholesterol (HDL-C), low-density lipoprotein cholesterol (LDL-C), and very low-density lipoprotein cholesterol (VLDL-C) in the plasma were segregated by density gradient ultracentrifugation (194,000× *g* at 10 °C for 3 h) determined by a Hitachi SP85G preparative ultracentrifuge (Tokyo, Japan), and then the HDL-C, LDL-C, and VLDL-C were recovered by tube slicing.

### 3.5. Measurement of Activities of Plasma Aspartate Aminotransferase (AST), Plasma Alanine Aminotransferase (ALT), Liver Superoxide Dismutase (SOD), and Liver Glutathione Peroxidase (GPx) and Content of Liver Lipid Peroxide (Thiobarbituric Acid Reactive Substances, TBARS)

The measurement of AST and ALT activities was performed by the enzymatic assay kits for AST and ALT (Randox Laboratories, Antrim, UK). A spectrophotometer (UV/VIS-7800, JASCO International, Tokyo, Japan) was used to detect the absorbance at 340 nm.

A SOD assay kit (Cayman Chemical, Ann Arbor, MI, USA) was used to detect the liver SOD activity. A VersaMax microplate (Molecular Device, San Jose, CA, USA) was used to measure the absorbance at 440 nm.

A glutathione peroxidase assay kit (Cayman Chemical, Ann Arbor, MI, USA) was used to measure the liver GPx activity. A VersaMax microplate (Molecular Device, San Jose, CA, USA) was used to analyze the absorbance at 340 nm.

The detection of TBARS was determined as previous described [40] with a modification. The reaction between thiobarbituric acid (TBA) and lipid peroxide product (malondialdehyde, MDA) to produce color was analyzed, and the lipid peroxide content in the liver was measured. The 1,1,3,3,-tetraethoxypropane (Sigma-Aldrich, St. Louis, MO, USA) was as a standard group and the physiological saline was as a blank group. A Hitachi U2800A spectrophotometer was used to detect the absorbance at 520 and 535 nm.

### 3.6. Detection of Lipolysis Rate and Lipoprotein Lipase (LPL) Activity

The detection of lipolysis rate was performed as previously described by Berger and Barnard [41]. Briefly, samples (0.2 g perirenal adipose tissues) were minced, and then mixed in 2 mL of 25 mM N-tris-(hydroxymethyl)methyl-2-aminoethanesulfonic acid buffer (pH 7.4) containing 1 μM isoproterenol at 37°C. A glycerol assay kit (Randox Laboratories, Antrim, UK) was used to measure the glycerol levels after 1, 2, and 3 h of incubation. A Hitachi U2800A spectrophotometer was used to detect the absorbance at 520 nm. An equation of micromoles of glycerol released per gram of adipose tissue per hour indicated the lipolysis rate.

The activity of LPL in the adipose tissues was analyzed as previously described by Kusunoki et al. [42]. Briefly, samples (0.1 g adipose tissues) were minced, and then mixed in Krebs–Ringer bicarbonate buffer (pH 7.4) containing 10 units/mL heparin for 60 min at 37 °C. The sample solution was then reacted with an equal volume of p-nitrophenyl butyrate (2 mM). A Hitachi U2800A spectrophotometer was used to measure the absorbance at 400 nm. The amount of p-nitrophenol formation over the 10 min incubation indicated the LPL activity.

### 3.7. Measurement of Activities of Hepatic Acetyl-CoA Carboxylase (ACC), Hepatic Fatty Acid Synthase (FAS), and Hepatic HMG-CoA Reductase (HMGCR)

The ACC activity was performed as previously described [43,44]. The samples and reagents (50 mM Tris-HCl buffer, 10 mM MgCl_2_, 10 mM potassium citrate, 3.75 mM glutathione, 12.5 mM KHCO_3_, 0.675 mM BSA, 0.125 mM acetyl-CoA, 3.75 mM ATP, liver cytosol preparations, and 10 mM NADPH) were mixed and reacted in 96-well microplates. A VersaMax microplate reader (Molecular Devices, San Jose, CA, USA) was used to measure the absorbance at 340 nm.

The analysis of FAS activity was determined as previously described [45,46]. The samples and reagents (0.2 M K_2_HPO_4_ buffer, 20 mM dithiothreitol (DTT), 0.25 mM acetyl-CoA, 60 mM EDTA–2Na, 0.39 mM malonyl-CoA, liver cytosol preparations, and 6 mM NADPH) were mixed and reacted in 96-well microplates. A VersaMax microplate reader (Molecular Devices, San Jose, CA, USA) was used to measure the absorbance at 340 nm.

The HMGCR activity was analyzed as previously described [46,47]. The liver microsome fraction was prepared as previously described by Krüner and Westernhagen [48]. The samples and reagents (0.2 M KCl, 0.16 M KH_2_PO_4_, 0.004 M EDTA, 0.01 M DTT, 0.1 mM HMG-CoA, liver microsomal preparation, and 0.2 mM NADPH) were mixed and reacted in 96-well microplates. A VersaMax microplate reader (Molecular Devices, San Jose, CA, USA) was used to detect the absorbance at 340 nm.

### 3.8. Analysis of Immunoblot

The methods of protein extraction and Western blot were performed as previously described by Chiu et al. [11]. Briefly, the radioimmunoprecipitation assay (RIPA) buffer with a cocktail of phosphatase and protease inhibitors (Thermo Fisher Scientific, Waltham, MA, USA) was used for liver protein extraction. A BCA protein assay kit (Thermo Fisher Scientific, Waltham, MA, USA) was used to measure protein concentration. For immunoblotting, 50–100 μg of proteins were separated through 8–12% SDS-PAGE gel and then transferred to polyvinylidene difluoride membranes (Bio-Rad, Hercules, CA, USA). After blocking using 5% non-fat dry milk solution for 1 h, membranes were probed with primary antibodies specific for phosphorylated AMP-activated protein kinase α (p-AMPKα; #2531; 1:1000), AMPKα (#2532; 1:1000) (Cell Signaling Technology, Danvers, MA, USA), PPARγ (sc-7273; 1:1000), and GAPDH (sc-47724; 1:1000) (Santa Cruz Biotechnology, Santa Cruz, CA, USA) at 4 °C overnight. The membranes were then probed with horseradish peroxidase-conjugated secondary antibodies (Cell Signaling Technology, Danvers, MA, USA). An Enhanced Chemiluminescence kit (Bio-Rad) was used to detect the target protein expression, which was exposed to a Fujifilm X-ray film (Tokyo, Japan). The densitometry of target protein was analyzed by the Image J 1.8 software (National Institutes of Health, Bethesda, MD, USA).

### 3.9. Analysis of Immunohistochemistry

The immunohistochemistry for 4-hydroxy-2-nonenal (4-HNE) staining was determined as previously described [49]. Briefly, liver tissue sections were sequentially deparaffinized, rehydrated, and antigen retrieval. Tissue section slides were blocked with bovine serum albumin and then incubated with primary antibody for 4-HNE (1:100; Abcam, Cambridge, MA, USA) overnight. Tissue section slides were reacted with biotin-conjugated secondary antibody and incubated in streptavidin-horseradish peroxidase solution. Finally, it was color-displayed using 3,3ʹ-diaminobenzidine tetrahydrochloride, a chromogenic substrate, to form the brown deposits.

### 3.10. Analysis of Quantitative Reverse Transcription Polymerase Chain Reaction (qRT-PCR)

The methods of total RNA extraction and qRT-PCR analysis in the small intestinal mucosa were determined as previously described by Chiu et al. [11]. The total RNA extraction was performed by a TRIzol kit (Life Technologies, Carlsbad, CA, USA). An ABI StepOne™ Real-Time PCR system and the software of StepOne 2.1 (Applied Biosystems, Foster City, CA, USA) were used to analyze qRT-PCR. The specific primers used are as follows: FABP2 (NM_013068.1; forward: TGAAAAGTTCATGGAGAAAATGG; reverse: CCTGTGTGATCGTCAGTTTCA), FATP4 (NM_001100706.1; forward: ATGACTGCCTCCCCCTCTAC; reverse: AGTCATGCCGTGGAGTACG), and GAPDH (NM_017008.4; forward: ATGACTCTACCCACGGCAAG; reverse: GGAAGATGGTGATGGGTTTC). GAPDH was used as an internal control. The levels of mRNA expression were normalized by the GAPDH level.

### 3.11. Examination of Liver Histology

The examination of liver histomorphology was performed as previously described [47]. The 5 μm-thick paraffin sections of liver tissues were stained with hematoxylin and eosin (H&E). An Olympus microscope equipped with a digital camera (U-LH100HG, BX53, Tokyo, Japan) was used to observe and photograph the stained images.

### 3.12. Statistics

All data are presented as mean ± standard deviation (S.D.). The statistical difference was evaluated by one-way analysis of variance (ANOVA) with a post-hoc test (Duncan’s multiple range test) using IBM SPSS statistics (version: 22.0; Armonk, NY, USA). It is considered as a significant difference when *p* is less than 0.05.

## 4. Conclusions

In this study, we found that both HC and CO supplementation in HF diet-induced obese rats for 12 weeks effectively reduced the increases in plasma and hepatic lipids and activated the AMPK activation, decreased the PPARγ protein expression, and decreased the lipogenesis-related enzyme (ACC, FAS, and HMGCR) activities in the livers. Both HC and CO could also reduce liver lipid biosynthesis, but HC had a better effect on improving liver lipid accumulation. However, HC mainly used its physical properties to increase lipid excretion and inhibit lipid absorption, while CO reduced lipid absorption by reducing the expression of *fabp2* and *fatp4* mRNA in the intestines. Moreover, HC, but not CO, supplementation could promote the liver antioxidant enzymes (GPx and SOD) activities and reduce the liver lipid peroxide production. The detailed mechanisms still need further investigation in the future.

Recently, it was mentioned that the application of CO as a dietary blend is still a controversial issue by regulatory authorities because of a lack of extensive safety data [50]. However, our recent study suggested that supplementation of 5% CO for 12 weeks did not induce lipid metabolism disorder and liver toxicity in normal rats [51]. A positive effect of CO supplementation on regulation of hepatic lipid metabolism has been reported in HF diet-fed mice with a long-term period of 4 months [52]. A long-term study may be needed to clarify the positive actions of CO in HF diet-induced obese rats.

## Figures and Tables

**Figure 1 ijms-22-01139-f001:**
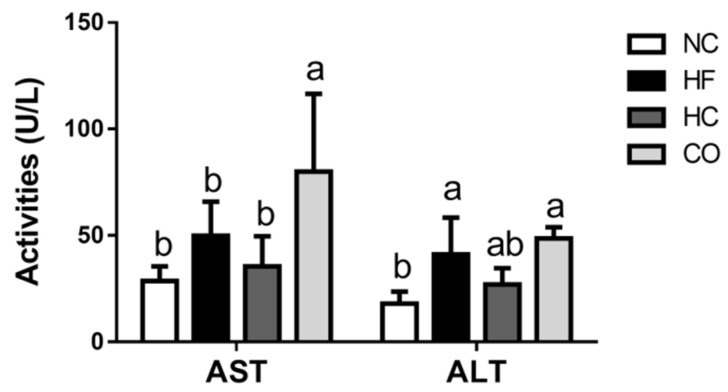
The changes in plasma aspartate aminotransferase (AST) and alanine aminotransferase (ALT) activities in rats fed with different experimental diets after 12 weeks. Data are expressed as the mean ± S.D. for each group (*n* = 8). Values with different superscript letters (a, b) are significantly different from each other (*p* < 0.05). NC: Normal control diet (chow diet). HF: High-fat diet (chow diet + 10% lard). HC: HF diet + 5% high-molecular-weight chitosan. CO: HF diet + 5% chitosan oligosaccharides.

**Figure 2 ijms-22-01139-f002:**
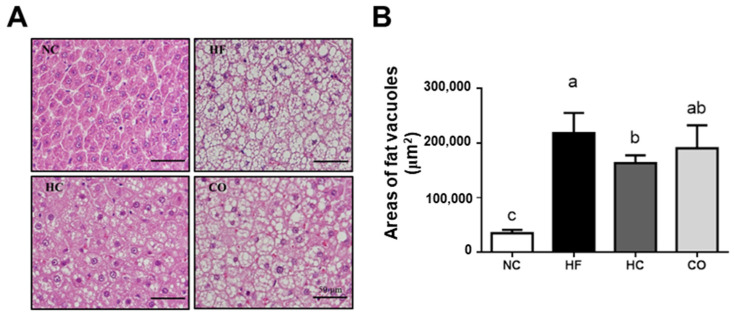
The changes in liver histological morphology in rats fed with different experimental diets for 12 weeks. (**A**) Liver tissue sections were stained with hematoxylin and eosin (H&E) staining. (**B**) Data are expressed as the mean ± S.D. for each group (*n* = 8). Values with different superscript letters (a, b, c) are significantly different from each other (*p* < 0.05). NC: Normal control diet (chow diet). HF: High-fat diet (chow diet + 10% lard). HC: HF diet + 5% high-molecular-weight chitosan. CO: HF diet + 5% chitosan oligosaccharides.

**Figure 3 ijms-22-01139-f003:**
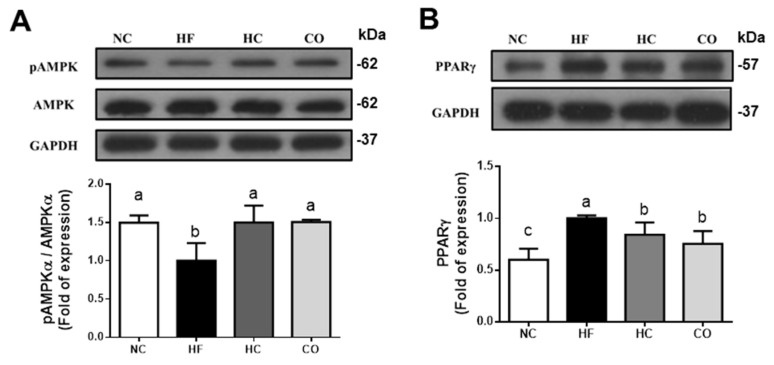
The changes in hepatic phosphorylated AMPKα. AMPKα (**A**), and PPAR-γ (**B**) protein expression in rats fed with different experimental diets for 12 weeks. Protein expression for pAMPKα, AMPKα, and PPAR-γ was determined by Western blotting. Densitometric analysis for protein levels corrected to each internal control was shown. Data are expressed as the mean ± S.D. for each group (*n* = 4–5). Values with different superscript letters (a, b, c) are significantly different from each other (*p* < 0.05). NC: Normal control diet (chow diet). HF: High-fat diet (chow diet + 10% lard). HC: HF diet + 5% high-molecular-weight chitosan. CO: HF diet + 5% chitosan oligosaccharides.

**Figure 4 ijms-22-01139-f004:**
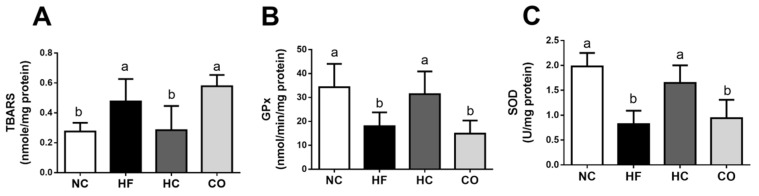
The changes in hepatic thiobarbituric acid reactive substances (TBARS) levels (**A**) and antioxidative enzymes (glutathione peroxidase (GPx) (**B**) and superoxide dismutase (SOD) (**C**) activities in rats fed with different experimental diets for 12 weeks. Data are expressed as the mean ± S.D. for each group (*n* = 8). Values with different superscript letters (a, b) are significantly different from each other (*p* < 0.05). NC: Normal control diet (chow diet). HF: High-fat diet (chow diet + 10% lard). HC: HF diet + 5% high-molecular-weight chitosan. CO: HF diet + 5% chitosan oligosaccharides.

**Figure 5 ijms-22-01139-f005:**
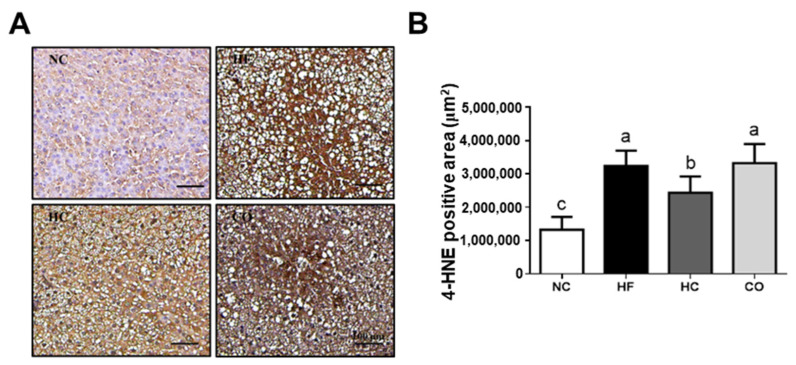
The changes in hepatic immunohistochemical staining for 4-hydroxy-2-nonenal (4-HNE) in rats fed with different experimental diets for 12 weeks. (**A**) Liver sections were immunohistochemically stained with 4-HNE. (**B**) Data are expressed as the mean ± S.D. for each group (*n* = 8). Values with different superscript letters (a, b, c) are significantly different from each other (*p* < 0.05). NC: Normal control diet (chow diet). HF: High-fat diet (chow diet + 10% lard). HC: HF diet + 5% high-molecular-weight chitosan. CO: HF diet + 5% chitosan oligosaccharides.

**Figure 6 ijms-22-01139-f006:**
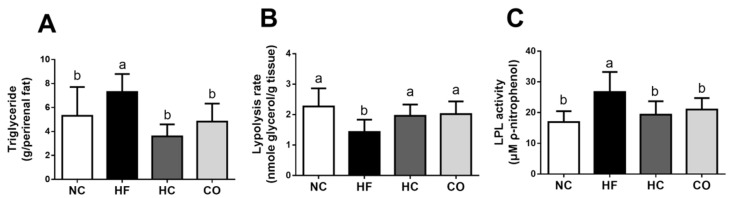
The changes in triglyceride level (**A**), lipolysis rate (**B**), and lipoprotein lipase (LPL) activity (**C**) in the perirenal adipose tissue of rats fed with different experimental diets for 12 weeks. Data are expressed as the mean ± S.D. for each group (*n* = 8). Values with different superscript letters (a, b) are significantly different from each other (*p* < 0.05). NC: Normal control diet (chow diet). HF: High-fat diet (chow diet + 10% lard). HC: HF diet + 5% high-molecular-weight chitosan. CO: HF diet + 5% chitosan oligosaccharides.

**Figure 7 ijms-22-01139-f007:**
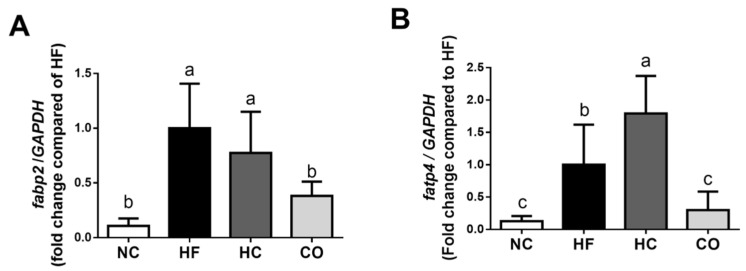
The changes in intestinal *fabp2* (**A**) and *fatp4* (**B**) mRNA expression in rats fed with different experimental diets for 12 weeks. Data are expressed as the mean ± S.D. for each group (*n* = 6). Values with different superscript letters (a, b, c) are significantly different from each other (*p* < 0.05). NC: Normal control diet (chow diet). HF: High-fat diet (chow diet + 10% lard). HC: HF diet + 5% high-molecular-weight chitosan. CO: HF diet + 5% chitosan oligosaccharides.

**Table 1 ijms-22-01139-t001:** The changes in body weight, food intake, and food utilization rate in rats fed with different experimental diets for 12 weeks.

Parameters	NC	HF	HC	CO
Initial body weight (g)	223.86 ± 5.17 ^a^	223.86 ± 7.92 ^a^	223.85 ± 6.11 ^a^	224.34 ± 5.09 ^a^
Final body weight (g)	527.07 ± 37.55 ^b^	602.45 ± 46.15 ^a^	537.28 ± 32.85 ^b^	557.56 ± 23.77 ^b^
Body weight gain (g)	304.01 ± 29.89 ^b^	377.99 ± 44.23 ^a^	314.14 ± 32.39 ^b^	331.49 ± 25.18 ^b^
Food intake (g/day)	27.07 ± 1.87 ^a^	25.79 ± 1.22 ^a,b^	24.12 ± 1.57 ^b^	24.89 ± 1.23 ^b^
Food efficiency (%)	11.22 ± 0.49 ^c^	14.61 ± 1.11 ^a^	13.00 ± 0.66 ^b^	13.33 ± 0.88 ^b^

Data are expressed as the mean ± SD for each group (*n* = 8). Values with different superscript letters (a, b, c) are significantly different from each other (*p* < 0.05). NC: Normal control diet (chow diet). HF: High-fat diet (chow diet + 10% lard). HC: HF diet + 5% high-molecular-weight chitosan. CO: HC diet + 5% chitosan oligosaccharides.

**Table 2 ijms-22-01139-t002:** The changes in body weight, food intake, and food utilization rate in rats fed with different experimental diets for 12 weeks.

Parameters	NC	HF	HC	CO
Liver weight (g)	15.18 ± 0.71 ^c^	28.31 ± 3.16 ^a^	21.57 ± 2.59 ^b^	27.19 ± 2.27 ^a^
Relative liver weight (g/100 g BW)	2.89 ± 0.10 ^c^	4.70 ± 0.31 ^a^	4.01 ± 0.35 ^b^	4.88 ± 0.45 ^a^
Perirenal fat (g)	10.56 ± 3.58 ^b^	13.85 ± 2.39 ^a^	8.57 ± 1.67 ^b^	10.57 ± 2.72 ^b^
Relative Perirenal fat weight (g/100 g BW)	2.00 ± 0.65 ^a,b^	2.30 ± 0.38 ^a^	1.59 ± 0.38 ^b^	1.88 ± 0.43 ^a,b^
Epididymal fat (g)	7.96 ± 1.93 ^b^	10.02 ± 1.62 ^a^	7.53 ± 1.69 ^b^	7.20 ± 1.80 ^b^
Relative Epididymal fat weight (g/100 g BW)	1.52 ± 0.36 ^a,b^	1.66 ± 0.22 ^a^	1.39 ± 0.29 ^a,b^	1.29 ± 0.28 ^b^
Total adipose tissue weight (g)	18.53 ± 5.38 ^b^	23.87 ± 3.85 ^a^	16.10 ± 3.27 ^b^	17.77 ± 4.10 ^b^
Relative adipose tissue weight (g/100 g BW)	3.52 ± 0.98 ^a,b^	3.97 ± 0.57 ^a^	2.98 ± 0.53 ^b^	3.17 ± 0.62 ^a,b^
Small intestine weight (g)	10.98 ± 0.94 ^a^	11.12 ± 1.05 ^a^	10.65 ± 1.42 ^a^	10.00 ± 1.07 ^a^
Small intestine length (cm)	135.63 ± 10.25 ^a^	132.00 ± 5.71 ^a^	131.56 ± 8.29 ^a^	132.44 ± 7.00 ^a^
Cecal weight (g)	8.14 ± 1.09 ^a^	6.37 ± 0.83 ^b^	8.46 ± 1.79 ^a^	5.51 ± 0.53 ^b^
Cecal content weight (g)	6.82 ± 1.03 ^a^	5.42 ± 0.80 ^b^	7.09 ± 1.72 ^a^	4.14 ± 0.44 ^c^

Data are expressed as the mean ± SD for each group (*n* = 8). Values with different superscript letters (a, b, c) are significantly different from each other (*p* < 0.05). NC: Normal control diet (chow diet). HF: High-fat diet (chow diet + 10% lard). HC: HF diet + 5% high-molecular-weight chitosan. CO: HF diet + 5% chitosan oligosaccharides.

**Table 3 ijms-22-01139-t003:** The changes in plasma biochemical indexes in rats fed with different experimental diets for 12 weeks.

Parameters	NC	HF	HC	CO
Total cholesterol (mg/dl)	56.57 ± 10.43 ^b^	74.10 ± 21.30 ^a^	44.67 ± 4.50 ^b^	53.81 ± 11.32 ^b^
HDL-C (mg/dl)	39.43 ± 6.37 ^a^	26.86 ± 5.48 ^b^	28.92 ± 4.50 ^b^	24.33 ± 3.38 ^b^
LDL-C + VLDL-C (mg/dl)	17.14 ± 5.23 ^c^	47.23 ± 17.03 ^a^	15.74 ± 3.09 ^c^	29.49 ± 8.50 ^b^
VLDL-C (mg/dl)	10.37 ± 2.70 ^c^	25.09 ± 4.64 ^a^	8.29 ± 2.91 ^c^	15.89 ± 3.50 ^b^
LDL-C (mg/dl)	6.77 ± 3.40 ^b^	22.14 ± 14.83 ^a^	7.45 ± 3.43 ^b^	13.60 ± 6.02 ^a,b^
TC / HDL-C (mg/dl)	1.43 ± 0.10 ^c^	2.75 ± 0.39 ^a^	1.56 ± 0.15 ^c^	2.19 ± 0.26 ^b^
HDL-C / (LDL-C + VLDL-C)	2.46 ± 0.63 ^a^	0.60 ± 0.13 ^b^	1.97 ± 0.76 ^a^	0.88 ± 0.21 ^b^
Triglyceride (mg/dl)	59.96 ± 12.41 ^a^	30.87 ± 7.31 ^b^	16.34 ± 4.01 ^c^	22.22 ± 5.10 ^b,c^
Blood glucose (mg/dl)	216.7 ± 7.64 ^b^	239.6 ± 4.102 ^a^	208.0 ± 6.122 ^b^	202.6 ± 9.021 ^b^

Data are expressed as the mean ± SD for each group (*n* = 8). Values with different superscript letters (a, b, c) are significantly different from each other (*p* < 0.05). NC: Normal control diet (chow diet). HF: High-fat diet (chow diet + 10% lard). HC: HF diet + 5% high-molecular-weight chitosan. CO: HF diet + 5% chitosan oligosaccharides.

**Table 4 ijms-22-01139-t004:** The changes in liver lipids and lipid biosynthesis-related enzyme activities in rats fed with different experimental diets for 12 weeks.

Parameters	NC	HF	HC	CO
Total cholesterol				
(mg/g liver)	4.26 ± 0.64 ^c^	110.04 ± 12.57 ^a^	84.10 ± 8.06 ^b^	86.25 ± 6.30 ^b^
(g/liver)	0.06 ± 0.01 ^d^	3.11 ± 0.43 ^a^	1.82 ± 0.31 ^c^	2.34 ± 0.26 ^b^
Triglyceride				
(mg/g liver)	20.58 ± 6.21 ^c^	80.64 ± 13.59 ^a^	62.32 ± 11.16 ^b^	64.98 ± 16.59 ^b^
(g/liver)	0.31 ± 0.09 ^c^	2.31 ± 0.31 ^a^	1.35 ± 0.31 ^b^	1.75 ± 0.41 ^b^
Acetyl-CoA carboxylase (ACC) (n mole NADPH/min/mg protein)	0.64 ± 0.13 ^c^	2.32 ± 0.24 ^a^	1.21 ± 0.19 ^b^	1.36 ± 0.34 ^b^
Fatty acid synthase (FAS) (n mole NADPH/min/mg protein)	0.33 ± 0.13 ^c^	1.57 ± 0.64 ^a^	0.77 ± 0.22 ^b^	0.78 ± 0.33 ^b^
HMG-CoA reductase (HMGCR) (n mole NADPH/min/mg protein)	0.29 ± 0.009 ^b^	0.56 ± 0.18 ^a^	0.35 ± 0.10 ^b^	0.39 ± 0.14 ^b^

Data are expressed as the mean ± SD for each group (*n* = 8). Values with different superscript letters (a, b, c, d) are significantly different from each other (*p* < 0.05). NC: Normal control diet (chow diet). HF: High-fat diet (chow diet + 10% lard). HC: HF diet + 5% high-molecular-weight chitosan. CO: HF diet + 5% chitosan oligosaccharides.

**Table 5 ijms-22-01139-t005:** The changes in fecal weight and levels of total cholesterol and triglyceride in rats fed with the different experimental diets after 12 weeks.

Parameters	NC	HF	HC	CO
Feces wet weight (g/day)	7.49 ± 1.70 ^a^	6.87 ± 0.96 ^a^	7.41 ± 0.69 ^a^	8.15 ± 1.25 ^a^
Feces dry weight (g/day)	5.80 ± 0.60 ^a^	5.49 ± 0.57 ^a^	5.79 ± 0.21 ^a^	5.96 ± 0.65 ^a^
Total cholesterol				
(mg/g feces)	6.02 ± 1.34 ^c^	10.56 ± 1.33 ^b^	12.21 ± 1.25 ^a^	10.84 ± 1.63 ^a,b^
(mg/day)	34.89 ± 8.08 ^c^	57.66 ± 6.63 ^b^	70.63 ± 7.23 ^a^	64.68 ± 12.05 ^a,b^
Triglyceride				
(mg/g feces)	5.69 ± 1.72 ^c^	6.68 ± 1.04 ^b,c^	8.69 ± 1.66 ^a^	7.64 ± 2.09 ^a,b^
(mg/day)	33.10 ± 10.49 ^c^	36.91 ± 8.12 ^b,c^	50.40 ± 10.30 ^a^	44.98 ± 10.40 ^a,b^
Bile acid (μmol/day)	25.5 ± 9.3 ^b^	25.3 ± 8.3 ^b^	39.1 ± 9.7 ^a^	25.9 ± 9.8 ^b^

Data are expressed as the mean ± SD for each group (*n* = 8). Values with different superscript letters (a, b, c) are significantly different from each other (*p* < 0.05). NC: Normal control diet (chow diet). HF: High-fat diet (chow diet + 10% lard). HC: HF diet + 5% high-molecular-weight chitosan. CO: HF diet + 5% chitosan oligosaccharides.

**Table 6 ijms-22-01139-t006:** Composition of experimental diets (%).

Ingredient (%)	NC	HF	HC	CO
Lard	―	10	10	10
Cholesterol	―	0.5	0.5	0.5
Cholic acid	―	0.1	0.1	0.1
Chitosan oligosaccharides ^1^	―	―	―	5
Chitosan ^2^	―	―	5	―
Chow diet	100	89.4	84.4	84.4
Total calories (kcal/100 g)	336.2	390.56	383.75	383.75
Carbohydrates (% kcal)	57.9	44.6	45.4	45.4
Protein (% kcal)	28.7	22.1	21.2	21.2

NC: Normal control diet (chow diet). HF: High-fat diet (chow diet + 10% lard). HC: HF diet + 5% high-molecular-weight chitosan. CO: HF diet + 5% chitosan oligosaccharides. ^1^ The average molecular weight (MW) and deacetylation degree (DD) of chitosan oligosaccharides are about 719 Dalton and 100%. ^2^ The average MW and DD of chitosan are about 7.4 × 10^5^ Dalton and 91%.

## Data Availability

The data presented in this study are available from the corresponding author upon reasonable request.

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
