# Peer review of "Effects and Mechanisms of Chitosan and ChitosanOligosaccharide on Hepatic Lipogenesis and Lipid Peroxidation, Adipose Lipolysis, and Intestinal Lipid Absorption in Rats with High-Fat Diet-Induced Obesity"

_ijms, 2021, doi:10.3390/ijms22031139_

Round 1
Reviewer 1 Report
In this study the authors have investigated how chitosan and its derivatives affect hepatic lipogenesis, lipid peroxidation and lipid absorption in rats with
high-fat diet-induced obesity. The study is well designed, well written and will be of great interest to a wide range of readers including scientist and clinicians. Briefly, the authors founds that chitosan supplementation reversed the effects of high fat diet i.e., increased triglyceride, decreased lipolysis rate, and increased lipoprotein lipase activity in the perirenal adipose tissue of the rats. Further they demonstrated that chitosan supplementation promoted antioxidant activity in the liver and protected against lipid peroxidation. Although the authors have done a great job with their experimental analysis, few more details or data are required to support their claims and make sure that the data is reproducible. See below for my comments.
- How did the authors determine the sample number for each experiment? Did they perform any power analysis to support this?
- What is the effect of chitosan supplementation of blood glucose levels in rats?
- Provide data on reactive oxygen species level and how chitosan supplementation affect this.
- Authors should justify as why no female rats were included in this study.
Author Response
Reviewer 1:
In this study the authors have investigated how chitosan and its derivatives affect hepatic lipogenesis, lipid peroxidation and lipid absorption in rats with high-fat diet-induced obesity. The study is well designed, well written and will be of great interest to a wide range of readers including scientist and clinicians. Briefly, the authors founds that chitosan supplementation reversed the effects of high fat diet i.e., increased triglyceride, decreased lipolysis rate, and increased lipoprotein lipase activity in the perirenal adipose tissue of the rats. Further they demonstrated that chitosan supplementation promoted antioxidant activity in the liver and protected against lipid peroxidation. Although the authors have done a great job with their experimental analysis, few more details or data are required to support their claims and make sure that the data is reproducible. See below for my comments.
Response:
We appreciate the reviewer's positive comment.
- How did the authors determine the sample number for each experiment? Did they perform any power analysis to support this?
Response: We appreciate the reviewer's comment. Based on the principles of the 3Rs (Replacement, Reduction and Refinement) for animal study, we used 12 rats per group in this study. The 4 rats/group used for the preliminary test to find the proper doses and experimental periods, and 8 rats/group used for formal experiments.
- What is the effect of chitosan supplementation of blood glucose levels in rats?
Response: We appreciate the reviewer's comment. We have added the data for blood glucose in Table 3 of this revised manuscript according to the suggestion of reviewer.
The blood glucose levels were increased in the HF diet-fed rats (P<0.05 vs NC group), which could be reversed by both HC and CO supplementation (P<0.05 vs HF diet-fed group) (Table 3).
- Provide data on reactive oxygen species level and how chitosan supplementation affect this.
Response: We appreciate the reviewer's comment. The detection of real reactive oxygen species (ROS) in tissues of animals is difficult. In this study, we used the alternative methods to detect the changes of oxidative stress, including the measurements of antioxidant enzymes GPx and SOD activities and production of lipid peroxidation for TBARS (lipid peroxide product malondialdehyde, MDA, levels) and 4-hydroxy-2-nonenal (4-HNE) in the livers. Our results showed that HC supplementation could significantly increase liver antioxidant enzymes GPx and SOD activities, and reduce the production of TBARS and 4-HNE in HF diet-fed rats.
- Authors should justify as why no female rats were included in this study.
Response: We appreciate the reviewer's comment. It is usually too difficult to detect the exact phase of the menstrual cycle with different plasma levels of sex hormones in female rats. Therefore, in this study, we used male rats that have proper responses. Nevertheless, it may need to investigate these experiments in female to see if the same results can be extended to female or not in the future. We added the description for discussing this issue in this revised manuscript according to the suggestion of reviewer.
Reviewer 2 Report
- Animal experiment. The no. of ethical approval is missing and should be added.
- Can Authors explain why they used deionized water in this study?
- Doses of chitosan and chitosan oligosaccharide used in this work should be justified.
- Material and methods. Description of detection of lipolysis. The source (depot) of fat tissue used in this assay should by provided by the Authors.
- Western blot. Dilution and cat. no. of all antibodies used by the Authors should be added.
- “The detection of total RNA extraction was performed by a TRIzol kit” Detection of total RNA by Trizol? This sentence is unclear and should be re-written. Trizol allows to isolate RNA but is not used for the detection.
- PCR primers: Accession no of all genes should be included.
- This is really unexpected that TG level in the circulation was decreased in the HF diet-fed. In my opinion the Authors should discuss this point in more detailed fashion. Is this observation supported by literature data?
- Western blot panels: molecular weight of all tested proteins should be added.
Author Response
Reviewer 2:
- Animal experiment. The no. of ethical approval is missing and should be added.
Response: We appreciate the reviewer's comment. We have added the approval number for animal experiment in the Methods of this revised manuscript according to the suggestion of reviewer. The approval number is 107038.
- Can Authors explain why they used deionized water in this study?
Response: We appreciate the reviewer's comment. We actually used water purified by reverse osmosis (RO), which removed most of the organics and ions present in the feed water. We revised the description for this issue in this revised manuscript according to the suggestion of reviewer.
- Doses of chitosan and chitosan oligosaccharide used in this work should be justified.
Response: We appreciate the reviewer's comment. We added a description for dose selection in this study according to the suggestion of reviewer. The doses for HC and CO were selected according to the previous studies [14,15,23] and our preliminary test.
- Material and methods. Description of detection of lipolysis. The source (depot) of fat tissue used in this assay should by provided by the Authors.
Response: We appreciate the reviewer's comment. The source of fat tissue used for lipolysis detection is the perirenal adipose tissue. We have added this description in the Methods of this revised manuscript according to the suggestion of reviewer.
- Western blot. Dilution and cat. no. of all antibodies used by the Authors should be added.
Response: We appreciate the reviewer's comment. We have added the dilution and cat. number of Western blot in the Methods of this revised manuscript according to the suggestion of reviewer. p-AMPKα: #2531, 1:1000; AMPKα: #2532, 1:1000; PPARγ: sc-7273, 1:1000, and β-actin: sc-517582, 1:1000.
- “The detection of total RNA extraction was performed by a TRIzol kit” Detection of total RNA by Trizol? This sentence is unclear and should be re-written. Trizol allows to isolate RNA but is not used for the detection.
Response: We appreciate the reviewer's comment. We have revised this description in the Methods of this revised manuscript according to the suggestion of reviewer. The total RNA extraction was performed by a TRIzol kit (Life Technologies, Carlsbad, CA, USA).
- PCR primers: Accession no of all genes should be included.
Response: We appreciate the reviewer's comment. We have added the accession number of all genes in the Methods of this revised manuscript according to the suggestion of reviewer. Accession number for FABP2: NM_013068.1; FATP4: NM_001100706.1; GAPDH: NM_017008.4.
- This is really unexpected that TG level in the circulation was decreased in the HF diet-fed. In my opinion the Authors should discuss this point in more detailed fashion. Is this observation supported by literature data?
Response: We appreciate the reviewer's comment. We have discussed this issue in this revised manuscript according to the suggestion of reviewer. This paradoxical decrease of plasma TG levels in HF diet group is exactly consistent with previous findings, which may be caused by the down-regulation of plasma microsomal angiopoietin-like 4 ( a suppressor of lipoprotein lipase) and hepatic TG transfer protein and apolipoprotein E (a VLDL-TG secretion enhancer), leading to the TG accumulation in the liver [11,23,24]. Moreover, a previous study has also found that the rate of secretion of VLDL by hepatocytes and the release of TG from the liver are decreased, when the TG levels in the liver exceed 10% [25]. These factors may cause the reduction of TG level in the plasma of the HF diet group.
- Western blot panels: molecular weight of all tested proteins should be added.
Response: We appreciate the reviewer's comment. We have added the molecular weight of all tested proteins in Figure 3 of this revised manuscript according to the suggestion of reviewer.